# WGA-M001, a Mixture of Total Extracts of *Tagetes erecta* and *Ocimum basilicum*, Synergistically Alleviates Cartilage Destruction by Inhibiting ERK and NF-κB Signaling

**DOI:** 10.3390/ijms242417459

**Published:** 2023-12-14

**Authors:** Eunjeong Oh, Hahyeong Jang, Subin Ok, Jiwon Eom, Hyunyong Lee, Sung Hun Kim, Jong Hwa Kim, Yu Mi Jeong, Kyeong Jin Kim, Seung Pil Yun, Hyung-Jun Kwon, In-Chul Lee, Ji-Young Park, Siyoung Yang

**Affiliations:** 1Department of Biological Sciences, Sungkyunkwan University, Suwon 16419, Republic of Korea; eunjeong8836@gmail.com (E.O.); jhh7071@g.skku.edu (H.J.); dhrtnqls55@g.skku.edu (S.O.); jessica8945@g.skku.edu (J.E.); 2Department of Biomedical Sciences, Ajou University Graduate School of Medicine, Suwon 16499, Republic of Korea; 3Wooree Green Science, Ansan 15409, Republic of Korea; bigfriend@wooree.co.kr (H.L.); shkim12@wooree.co.kr (S.H.K.); kimjh1012@wooree.co.kr (J.H.K.); jym1@wooree.co.kr (Y.M.J.); kjkim@wooree.co.kr (K.J.K.); 4Department of Horticulture, Chonnam National University, Gwangju 61186, Republic of Korea; 5Department of Pharmacology, Institute of Health Sciences, Gyeongsang National University College of Medicine, Jinju 52727, Republic of Korea; spyun@gnu.ac.kr; 6Center for Companion Animal New Drug Development, Jeonbuk Branch, Korea Institute of Toxicology, Jeongeup 53212, Republic of Korea; hjkwon@kribb.re.kr (H.-J.K.); leeic@kribb.re.kr (I.-C.L.); loveme@kribb.re.kr (J.-Y.P.)

**Keywords:** osteoarthritis, *Tagetes erecta*, *Ocimum basilicum*, WGA-M001

## Abstract

*Tagetes erecta* and *Ocimum basilicum* are medicinal plants that exhibit anti-inflammatory effects against various diseases. However, their individual and combined effects on osteoarthritis (OA) are unknown. Herein, we aimed to demonstrate the effects of *T. erecta, O. basilicum*, and their mixture, WGA-M001, on OA pathogenesis. The administration of total extracts of *T. erecta* and *O. basilicum* reduced cartilage degradation and inflammation without causing cytotoxicity. Although WGA-M001 contained lower concentrations of the individual extracts, it strongly inhibited the expression of pathogenic factors. In vivo OA studies also supported that WGA-M001 had protective effects against cartilage destruction at lower doses than those of *T. erecta* and *O. basilicum*. Moreover, its effects were stronger than those observed using *Boswellia* and *Perna canaliculus.* WGA-M001 effectively inhibited the interleukin (IL)-1β-induced nuclear factor kappa-light-chain-enhancer of the activated B cell (NF-κB) pathway and ERK phosphorylation. Furthermore, RNA-sequence analysis also showed that WGA-M001 decreased the expression of genes related to the IL-1β-induced NF-κB and ERK signaling pathways. Therefore, WGA-M001 is more effective than the single total extracts of *T. erecta* and *O. basilicum* in attenuating OA progression by regulating ERK and NF-κB signaling. Our results open new possibilities for WGA-M001 as a potential therapeutic agent for OA treatment.

## 1. Introduction

Degenerative joint diseases, also known as osteoarthritis (OA), are significant and have become an increasingly prevalent health concern in the aging world population [1]. OA is a progressive condition characterized by gradual deterioration of the protective cartilage present within joints, resulting in damage to the underlying bones and ligaments [2]. The degeneration causes inflammation and pain, making it a critical disease to manage as the number of affected patients continue to increase [3]. Cartilage is a specialized connective tissue that plays a crucial role in supporting and cushioning joints, and provides a smooth surface for frictionless movement [4]. It is composed of chondrocytes, extracellular matrix, and molecules produced by the chondrocytes, such as collagen, proteoglycans, and glycosaminoglycans [5,6]. In OA, interleukin (IL)-1β binds to specific receptors on the surface of chondrocytes, activating intracellular signaling pathways [7], which lead to the upregulation of matrix metalloproteinases (MMPs) and aggrecanases—enzymes responsible for breaking down the extracellular matrix of cartilage, including collagen and proteoglycans [8,9,10]. Moreover, the upregulation of COX-2 and IL-6 enhances inflammatory responses [11,12], resulting in an imbalance between matrix degradation and synthesis, which exacerbates cartilage erosion [13,14].

In addition to surgery, drug therapy is commonly used to treat OA [15], which primarily includes medications with analgesic and anti-inflammatory properties [15]. However, a definitive cure for OA has not been developed [16]. Non-steroidal anti-inflammatory drugs are the most widely used class of drugs for OA treatment [17]. However, these drugs have several potential side effects, which make their long-term use challenging [18]. Thus, there is considerable interest in using natural products with chondroprotective and anti-inflammatory activities as functional foods that can protect the cartilage from deterioration [19].

The development and progression of OA occur via two important signaling pathways: the mitogen-activated protein kinase (MAPK) and nuclear factor kappa-light-chain-enhancer of activated B cell (NF-κB) signaling pathways [20]. The MAPK signaling pathway involves a serine/threonine protein kinase that is widely present in eukaryotic cells and can be activated by IL-1β [21]. Upon activation, MAPK transmits extracellular signals to the nucleus to regulate the activity of several transcription factors, resulting in the altered expression of genes encoding MMPs, COX-2, and IL-6 [22,23,24]. NF-κB is a well-known inflammatory signaling pathway that is also activated by IL-1β [25,26]. Activation of this pathway causes phosphorylation of p65 and IκBα, which results in the translocation of NF-κB into the nucleus, where it subsequently binds to specific sequences within the promoter regions of the target genes encoding pro-inflammatory mediators like COX-2, MMPs, and IL-6 [27]. Both the MAPK and NF-κB signaling pathways contribute to disease pathogenesis in OA by promoting inflammation, cartilage breakdown, and other pathological changes within joints [21,26]. Therefore, modulation of MAPK and NF-κB signaling may prove beneficial in the treatment of OA.

*Tagetes erecta*, commonly known as marigold, has been used for medicinal purposes since pre-Hispanic times [28]. It is used to treat digestive disorders such as abdominal pain, diarrhea, vomiting, and indigestion, as well as to treat inflammatory conditions such as bronchitis [29]. *Ocimum basilicum*, also known as basil, has diverse applications in traditional medicines, pharmaceuticals, cosmetics, and as a functional food [30]. It contains phenolic and flavonoid compounds that exhibit several pharmacological effects, such as antioxidant, anti-inflammatory, and analgesic effects, making it effective against chronic inflammation and immune-related diseases [31]. Although both *T. erecta* and *O. basilicum* are medicinal plants commonly used as dietary supplements [30,32], their function in OA is largely unknown. Herein, we demonstrate that WGA-M001, a mixture of *T. erecta* and *O. basilicum* extracts, strongly attenuates OA development through the inhibition of gene signatures related to ERK and NF-kB pathway activation, compared to the attenuation provided by a single extract of *T. erecta* or *O. basilicum*. Moreover, the effect of WGA-M001 strongly suggests the possibility of OA treatment in vitro and in vivo.

## 2. Results

### 2.1. Therapeutic Effects of T. erecta and O. basilicum under In Vitro Conditions Mimicking OA

*T. erecta* and *O. basilicum* were tested on mouse articular chondrocytes for cytotoxicity analysis. Mouse articular chondrocytes were treated with various concentrations of *T. erecta* or *O. basilicum* extract for 24 h. Through lactate dehydrogenase (LDH) assay, it was found that concentrations up to 200 µg/mL did not affect chondrocyte viability (Appendix A). Thus, use of up to 200 µg/mL of *T. erecta* and *O. basilicum* extract was suggested to be safe on chondrocytes. The inhibitory effect of *T. erecta* and *O. basilicum* on pathogenic factor expression in IL-1β-treated cells in vitro was further analyzed. *T. erecta* inhibited the transcription (Figure 1A,B) and protein levels (Figure 1C–E) of MMP3, MMP13, COX-2, and IL-6 in a dose-dependent manner. Moreover, compared to *T. erecta*, *O. basilicum* had a similar effect on the IL-1 β treated chondrocytes (Figure 1F–I). Although *T. erecta* and *O. basilicum* showed similar effects in inhibiting catabolic factor expression, treatment with *O. basilicum* had a stronger inhibitory effect on pathogenic factor expression than that associated with treatment with *T. erecta*. Thus, the data suggested that an appropriate proportion of the mixture can strongly inhibit OA pathogenesis by suppressing the expression of catabolic factors. The therapeutic effects of WGA-M001 were then examined using a mixture with an appropriate ratio of *T. erecta* to *O. basilicum*, under in vitro conditions that mimic OA. We also demonstrated that WGA-M001 had no cytotoxic effects on mouse articular chondrocytes as observed with respect to *T. erecta* and *O. basilicum* (Appendix A).

### 2.2. WGA-M001 Effectively Attenuates Catabolic Factor Expression

WGA-M001 (100 μg/mL and 200 μg/mL) was used with IL-1β as co-treatment of mouse chondrocytes. The expression of catabolic factors was examined using biochemical analyses. We demonstrated that the transcription levels of MMP3, MMP13, COX-2, and IL-6 were suppressed in IL-1β-treated conditions that mimic OA (Figure 2A,B). The expression level of MMP3, MMP13, COX-2, and IL-6 increased with the IL-1β treatment, compared to the levels in the control, whereas IL-1β induced catabolic factors decreased in the presence of WGA-M001 in a dose-dependent manner. Moreover, the decrease in the level of catabolic factor caused by WGA-M001 (200 μg/mL) was greater than the decrease associated with the same concentration of single extract *T. erecta* or *O. basilicum* in chondrocytes. This data collectively suggested that WGA-M001 has an inhibitory effect on catabolic factor expression, compared to *T. erecta* and *O. basilicum* treatment. The similarity between the in vitro OA mimic phenotypes and in vivo OA mouse experiments was also studied.

### 2.3. T. erecta and O. basilicum Mixtures Show a Synergistical Effect in the DMM-Induced OA Mouse Model

To verify the in vivo functions of WGA-M001, *T. erecta*, and *O. basilicum*, we destabilized the medial meniscus (DMM)-induced OA according to the schedule shown in Appendix A. The DMM-induced OA mouse model is a highly representative in vivo model for OA [33,34]. WGA-M001, *T. erecta*, or *O. basilicum* was orally administered to mice with DMM-induced OA. To determine the severity of OA development, the cartilage destruction using subchondral bone plate (SBP) thickness and osteophyte maturity was examined [35]. SBP plays an important role in articular cartilage metabolism [36]. Osteophyte maturity indicates the amount of bone tissue in osteophytes [37]. OA is a whole-joint disease characterized by increased cartilage destruction, osteophyte formation, and SBP thickness [38].

As shown by Safranin O staining in Figure 3A, WGA-M001 had a suppression of cartilage destruction compared to the effect of the oral administration of phosphate-buffered saline (PBS) as a control in DMM-induced OA mice. Oral administration of a small amount of WGA-M001 (up to 100 mg/kg) provided protection against cartilage destruction, whereas high doses (500 mg/kg) of *T. erecta* and *O. basilicum* showed suppression of cartilage destruction. This shows that the amount of WGA-M001 oral administration is less than that of the *T. erecta* and *O. basilicum* extracts for suppressing cartilage destruction. Therefore, WGA-M001 is more attractive for OA treatment than the use of single extracts. The OARSI grade, SBP thickness, and osteophyte maturity showed results similar to the cartilage destruction phenotype observed with Safranin O staining (Figure 3B–D). *Boswellia* and *Perna canaliculus* were used as positive controls. *Boswellia* and *P. canaliculus* are health functional foods commonly used for treatment of OA [39,40,41]. For centuries, extracts from *Boswellia* have held a significant place in traditional folk medicine [42], gaining popularity among consumers as a remedy for arthritis [43]. As shown in Figure 3, WGA-M001 had a protective effect against cartilage destruction, compared to *Boswellia* and *P. canaliculus*. This result demonstrated that a mixture of *T. erecta* and *O. basilicum* showed a combined effect, resulting in cartilage recovery at lower concentrations compared to the recovery associated with each substance alone.

### 2.4. WGA-M001 Attenuates OA Pathogenesis through Effective Gene Regulation with Respect to ERK and NF-κB Signaling

To demonstrate that we demonstrated that WGA-M001 has an inhibitory effect on catabolic factor expression, compared to *T. erecta* and *O. basilicum* treatment through inhibition of ERK and NF-κB pathway, RNA-seq analysis was performed to examine the regulation of the signaling pathways. NF-κB and MAPK (ERK, Jun N-terminal kinase (JNK), and p38) signaling pathways were primarily associated with the IL-1β signaling pathway [7]. In silico analysis was performed to determine the pathways through which *T. erecta*, *O. basilicum,* and WGA-M001 regulate OA. Alterations in the expression patterns of genes related to each pathway mediated by WGA-M001, *T. erecta*, and *O. basilicum* were analyzed. A list of genes involved in each signaling pathway was derived using Ingenuity Pathway Analysis (IPA) [44]. Alterations in expression patterns are presented as a heatmap (Figure 4A). Only genes related to each pathway whose expression was upregulated by IL-1β have been presented. Moreover, the expression patterns of WGA-M001, *T. erecta*, and *O. basilicum* were analyzed. Among the genes whose expression increased in the presence of IL-1β, we analyzed the number of genes whose expression increased to a greater degree following co-treatment with WGA-M001 compared to the increase associated with treatment with *T. erecta* and *O. basilicum* (Figure 4A). A total of 57 out of 72 genes related to the NF-κB signaling pathway and 59 out of 81 genes associated with ERK signaling were downregulated by WGA-M001, compared to the expression levels associated with other substances (Figure 4A). The results suggested that WGA-M001 blocked NF-κB and ERK signaling pathways, but not the other signaling pathway. We also investigated whether protein expression levels correlated with the heatmap outcomes (Figure 4B). As shown by the heatmap, the protein level of NF-κB and pERK was downregulated in the presence of WGA-M001 to a larger extent than the downregulation of the others (Figure 4C).

### 2.5. WGA-M001 Contains Quercetagetin-7-O-β-D-Glucopyranoside, Chicoric Acid, and Rosmarinic Acid

High-performance liquid chromatography (HPLC) experiments were performed to confirm the presence of phytochemical compounds in WGA-M001 (Figure 5). Comparative analysis using HPLC with standard compounds showed that Quercetagetin-7-O-β-D-glucopyranoside, Chicoric acid, and Rosmarinic acid were the most abundant in the sample treated with WGA-M001, and were detected at retention times of 16.6, 27.1, and 31.5 min, respectively.

Thus, these data indicate that the effect of WGA-M001 is a synergistical effects of *T. erecta* and *O. basilicum*, and inhibits catabolic factor expression through downregulation of the NF-κB and ERK signaling pathways (Figure 6).

## 3. Discussion

Owing to an increase in the elderly population and prevalence of obesity, the number of patients with OA is continuously increasing [45]. Symptoms such as joint pain, stiffness, swelling, and tenderness significantly affect the patient’s quality of life [46,47]. Therefore, there is a significant need for preventing the onset of OA and alleviating its symptoms. It is well known that the pro-inflammatory cytokine IL-1β increases in the cartilage of patients with OA. IL-1β activates downstream signaling pathways like NF-κB and MAPK in the cartilage [48]. NF-kB is an important transcription factor involved in immunity, inflammation, cell differentiation, and cell survival [49]. In OA, IL-1β induces NF-κB activation through IκB degradation and p65 phosphorylation, leading to the upregulation of catabolic factors like MMP3 and MMP13, as well as an increase in inflammation-related molecules such as COX-2 and IL-6 [49]. Similarly, the MAPK signaling pathway, consisting of three families (ERK, p38, and JNK), is involved in cell proliferation, inflammation, apoptosis, and differentiation [50]. Activation of this pathway leads to the upregulation of catabolic factors such as MMP3 and MMP13 as well as an increase in inflammation-related molecules such as COX-2 and IL-6 [23,51,52].

Accordingly, many researchers continue their efforts to identify safe and sustainable drugs that can inhibit these molecules, and there is growing interest in natural products for this purpose [19]. *T. erecta* has been studied for its ability to regulate NF-κB and p65 signaling in gastric cancer, induce anti-inflammatory responses, and reduce IL-6 levels [53]. In chronic inflammatory conditions such as ulcerative colitis, *T. erecta* has shown effectiveness by reducing TNF-α and IL-6 levels [29]. Similarly, *O. basilicum* has been studied for its ability to regulate NF-κB signaling in adipocytes, inducing anti-inflammatory responses by controlling IL-6, IL-1β, and TNF-α [54]. Moreover, when *T. erecta* and *O. basilicum* were assessed via HPLC analysis, Quercetagetin-7-O-β-D-glucopyranoside was mainly present in *T. erecta*, whereas Chicoric acid and Rosmarinic acid were found in *O. basilicum* (data were not shown). However, WGA-M001 contained Quercetagetin-7-O-β-D-glucopyranoside, Chicoric acid, and Rosmarinic acid. The ratio of *T. erecta* to *O. basilicum* was 10:1. This combination may show a stronger inhibitory effect on WGA-M001 compared to that of *T. erecta* and *O. basilicum* total extracts. The HPLC data for WGA-M001 showed several peaks in a time-dependent manner. Although we have not characterized all the single compounds, they could be valuable single compounds with suppression of catabolic factor expression and cartilage destruction. In the near future, we will isolate each compound and characterize its inhibition effect on catabolic factor expression and cartilage destruction. These findings suggested that the two substances may have significant effects on OA.

In this study, the effect of *T. erecta*, *O. basilicum*, or a combination of the two, WGA-M001, to alleviate the pathogenesis of OA was investigated. Although *T. erecta*, *O. basilicum*, and WGA-M001 suppressed catabolic factor expression without cytotoxicity in chondrocytes, WGA-M001 had an additional inhibitory effect with respect to the catabolic factor expression through inhibition of gene regulation related to the ERK and NF-κB signaling pathway, compared to the effect of *T. erecta* and *O. basilicum.* Moreover, in DMM-induced OA mice, oral administration of WGA-M001 at doses up to 100 mg/kg effectively inhibited cartilage destruction compared to that associated with a high dose (500 mg/kg) of *T. erecta* or *O. basilicum*. Recent research has shown that *Boswellia* and *P. canaliculus* are well-known OA-alleviating substances [41,55,56]. However, our study demonstrated that WGA-M001 had a stronger protective effect against OA development compared to the effect of the positive controls. Therefore, our in vitro, in vivo, and in silico analyses collectively suggest that WGA-M001, a mixture of both *T. erecta* and *O. basilicum*, may become a new treatment approach instead of *Boswellia* and *P. canaliculus* for OA and cartilage protection.

## 4. Materials and Methods

### 4.1. Sample Treatment and Reagents

*T. erecta*, *O. basilicum*, and WGA-M001 samples were supplied by the Wooree Green Science Corporation (Ansan, Korea). Dried *T. erecta* and *O. basilicum* was extracted using 50% EtOH for 24 h at 25 °C. The extracts were filtered using a Buchner funnel and concentrated using a rotary evaporator to obtain a 50% EtOH extraction powder. The 50% ethanol extract was fractionated using ethyl acetate and concentrated using a rotary evaporator. The dried samples were then stored at −80 °C in the refrigerator until further use. WGA-M001 is a mixture of two herbal extracts, i.e., *T. erecta* 50% ethanol extract powder and *O. basilicum* 50% ethanol extract powder in a ratio of 10:1. Samples were dissolved in dimethyl sulfoxide (DMSO) and boiled at 55 °C for 15 min. Human IL-1β recombinant protein (Z02922-10; GenScript, Piscataway, NJ, USA) was dissolved in PBS with 0.1% BSA. Twelve hours prior to cell harvesting, the samples (50, 100, and 200 μg/mL) were used along with IL-1β for co-treatment of mouse articular chondrocytes.

### 4.2. Primary Culture of Mouse Chondrocytes and Experimental Animals

Articular chondrocytes were isolated from the femoral condyles and tibial plateaus of 5-day-old ICR mice (DBL; Chungcheongbuk-do, Eumseong, Republic of Korea). Chondrocytes were isolated from cartilage tissues using Type II collagenase as previously described [57,58], and cultured in DMEM supplemented with 10% fetal bovine serum (FBS; Capricon, Ebsdorfergrund, Germany) and 1% penicillin/streptomycin. Male C57BL/6J mice (10 weeks old) were purchased from DBL (Chungcheongbuk-do, Eumseong, Republic of Korea). All in vitro experiments were performed at least four times. All the experimental protocols were approved by the Animal Care and Use Committee of the University of Ajou (approval number: IACUC 2016-0041). The mice were housed at a temperature of 25 °C with a 12 h light/dark cycle. All mouse in vivo experiments were performed according to previously described protocols [59].

### 4.3. LDH Assay

The LDH assay was performed to measure the cytotoxicity of *T. erecta*, *O. basilicum,* and WGA-M001. Approximately 24 h before cell harvest, the medium was changed to 0% FBS medium and the cells were treated with *T. erecta*, *O. basilicum*, or WGA-M001 at 50, 100, and 200 μg/mL each. The LDH assay was performed using chondrocyte supernatants. To measure LDH activity, cytotoxicity was normalized to that of the untreated samples (0% cytotoxicity) and samples treated with Triton X-100 (100% cytotoxicity). Cell viability (%) was calculated using the formula 100 − (Sample OD − untreated sample OD)/(Triton X-100 OD − untreated sample OD) × 100. Absorbance was measured at 490 nm using a microplate reader (VICTOR X3; PerkinElmer, Waltham, MA, USA).

### 4.4. RT-PCR and Quantitative RT-PCR (qRT-PCR)

Total RNA was isolated from mouse chondrocytes using TRIzol reagent (Molecular Research Center Inc., Cincinnati, OH, USA). Total chondrocyte RNA was reverse transcribed and cDNA was synthesized using the ImProm-IITM Reverse Transcriptase kit (A3803; Promega, Madison, WI, USA) and oligo-dT primers. Primer and PCR conditions (annealing temperature and cycles) used for each gene are described as follows: mouse MMP3 (5′-CTGTGTGTGGTTGTGTGCTCATCCTAC-3′ and 5′-GGCAAATCCGGTGTATAATTCACAATC-3′; 58 °C, 21 cycles), mouse MMP13 (5′-TGATGGACCTTCTGGTCTTCTGGC-3′ and 5′-CATCCACATGGTTGGGAAGTTCTG-3′; 58 °C, 21 cycles), mouse COX2 (5′-GGTCTGGTGCCTGGTCTGATGAT-3′ and 5′-GTCCTTTCAAGGAGAATGGTGC-3′; 64 °C, 26 cycles), IL-6 (5′-ACCACTCCCAACAGACCTGTCTATACC-3′ and 5′-CTCCTTCTGTGACTCCAGCTTATCTGTTAG-3′; 60 °C, 28 cycles), and mouse glyceraldehyde 3-phosphate dehydrogenase (GAPDH) (5′-TCACTGCCACCCAGAAGAC-3′ and 5′-TGTAGGCCATGAGGTCCAC-3; 58 °C, 21 cycles). To quantify the transcription levels of the genes, qRT-PCR was performed using SYBR Premix Ex Taq (Takara Bio, Shiga, Japan) and the transcription levels were normalized to those of GAPDH.

### 4.5. Protein Extraction, Western Blotting, and IL-6 ELISA Assay

Mouse chondrocyte proteins were extracted using lysis buffer (150 mM NaCl, 50 mM Tris, 0.2% sodium dodecyl sulfate, 1% NP-40, and 5 mM NaF) supplemented with a protease inhibitor cocktail (Roche, Madison, WI, USA). Trichloroacetic acid precipitation was performed to extract MMP3 and MMP13 from the chondrocyte culture medium. Proteins were separated by size on an acrylamide gel and transferred onto a PVDF membrane. Primary antibodies used were as follows: mouse anti-ERK (sc-514302; Santa Cruz, Dallas, TX, USA), rabbit anti-MMP3 (ab52915; Abcam, Cambridge, UK), rabbit anti-MMP13 (ab51072; Abcam), rabbit anti-COX-2 (ab52237; Abcam), mouse anti-IκB (9242; Cell Signaling Technology (CST), Danvers, MA, USA), mouse anti-p65 (#6956; CST), mouse anti-pp65 (#13346; CST), mouse anti-p38 (#9212; CST), mouse anti-pp38 (#9215S; CST), mouse anti-c-JNK (#9252S; CST), mouse anti-pJNK (#9251S; CST), and mouse anti-pERK (#9101S; CST). Protein levels were normalized to the ERK levels. The relative intensities of the bands were quantified by densitometric analysis using ImageJ software v1.60. IL-6 production was quantified using a mouse IL-6 ELISA kit (Koma Biotech, Seoul, South Korea), according to the manufacturer’s instructions. Absorbance for IL-6 was measured at 450 nm using a microplate reader (VICTOR X3; PerkinElmer, Waltham, USA) [33,34,59].

### 4.6. In Silico Analysis

*T. erecta* (200 μg/mL), *O. basilicum* (200 μg/mL), or WGA-M001 (200 μg/mL) along with IL-1β (1 ng/mL) was used as co-treatment of the mouse chondrocytes. For in silico analyses, total RNA was isolated using the TRIzol reagent (Molecular Research Center, Cincinnati, OH, USA). The result of RNA sequencing was merged with a gene list related to ERK, NF-κB, JNK, and p38 signaling pathway from IPA (http://www.ingenuity.com, accessed on 10 August 2023) and expression pattern was analyzed using the heatmap analysis software (ver. 4.3.2; Broad Institute, MIT, Cambridge, MA, USA). All in silico analyses were performed as previously described [44].

### 4.7. DMM-Induced OA Mouse Model and Oral Administration

For the experimental OA model, 10-week-old male C57BL/6 mice underwent DMM surgery using a previously described protocol [33,34]. Each experiment was conducted independently at least five times. To perform DMM surgery, the fat pad was removed and the medial meniscus tibial ligament was incised. This method increased the mechanical stress in the knee joints of mice and led to cartilage degradation and osteoarthritis-like changes. After DMM surgery, mice were randomly divided into 10 groups: PBS, *T. erecta* (200 and 500 mg/kg), *O. basilicum* (200 and 500 mg/kg), and WGA-M001 (50, 100, and 200 mg/kg). Mice were orally administered PBS, *T. erecta*, *O. basilicum,* or WGA-M001 at different concentrations for 10 weeks [57,60]. DMM-induced OA mice were orally administered *Boswellia* or *P. canaliculus* daily as positive controls.

### 4.8. Histological Analysis of Cartilage Destruction

DMM-induced OA mouse cartilage samples were obtained 10 weeks after DMM surgery, fixed in 4% paraformaldehyde overnight, and decalcified in 0.5 M EDTA for 2 weeks. The samples were then embedded in paraffin and the paraffin blocks were cut into sections of 5 μm thickness each. The sections were stained with hematoxylin, Safranin O, and fast green [33,34]. The histological results were scored using the OARSI (Osteoarthritis Research Society International) grade, osteophyte maturity grade, and subchondral bone plate thickness method. A representative Safranin O staining image was selected from serial sections displaying the most advanced lesions. Three observers were blinded to the experimental group which evaluated the OARSI scores. The OARSI scoring system comprises seven grades (0–6). Grade 0 indicates a joint with no signs of osteoarthritis. Grades 1–6 signifies different levels of abnormality in the cartilage matrix and the presence of cells in the superficial zone. Specifically, they correspond to surface discontinuity, vertical fissures, erosion, denudation, and deformation [44,61].

### 4.9. HPLC Analysis of WGA-M001

HPLC analysis of WGA-M001 for chemical characterization and quantitative analysis of the selected compounds (Quercetagetin-7-*O*-*β*-D-glucopyranoside, Chicoric acid and Rosmarinic acid) were carried out on an Agilent 1260 Infinity II with diode array detector (Agilent Technologies, Santa Clara, CA, USA). The sample containing WGA-M001 was dissolved in 50% EtOH and filtered through a 0.45 μm nylon membrane filter. The components were separated using a Capcell Pak MG II C18 column (4.6 × 250 mm, 5 μm), with a column temperature of 40 °C. The mobile phase consisted of (A) 0.3% phosphoric acid in water and (B) acetonitrile at a flow rate of 1 mL/min with gradient elution as follows: 0–20 min, 5–25% B; 20–35 min, 25–30% B; 35–50 min, 30–50% B. UV detection was performed at 330 nm and the injection volume was 10 μL. The measurements were repeated thrice for each sample. The selected components were identified by comparing their respective retention times and UV/Vis spectra with those of standard compounds. Quantification of the individual components in WGA-M001 was performed by peak area measurements using a calibration curve of each reference standard with a range of 6.25–100 μg/mL. The detector response was linear over the tested concentration range. The regression coefficients (*R*^2^) of the standard compounds were greater than 0.999.

### 4.10. Statistical Analysis

Statistical comparisons were performed using a series of tests. Normality was assessed using the Shapiro–Wilk test and homogeneity of variance was checked using Levene’s test [59]. Parametric data involving two groups were analyzed using a two-tailed independent t-test, whereas comparisons among three or more groups were performed using one-way analysis of variance with Dunnett’s multiple comparisons test. Non-parametric data for the two groups were assessed using the Mann–Whitney U test, whereas comparisons involving multiple groups were analyzed using either the Kruskal–Wallis test or Friedman test, followed by Dunn’s multiple comparisons test for post-hoc analysis. Results are presented as mean ± standard deviation and statistical significance was defined as *p* < 0.05. Statistical analyses were performed using GraphPad Prism 7 software (GraphPad, San Diego, CA, USA).

## Figures and Tables

**Figure 1 ijms-24-17459-f001:**
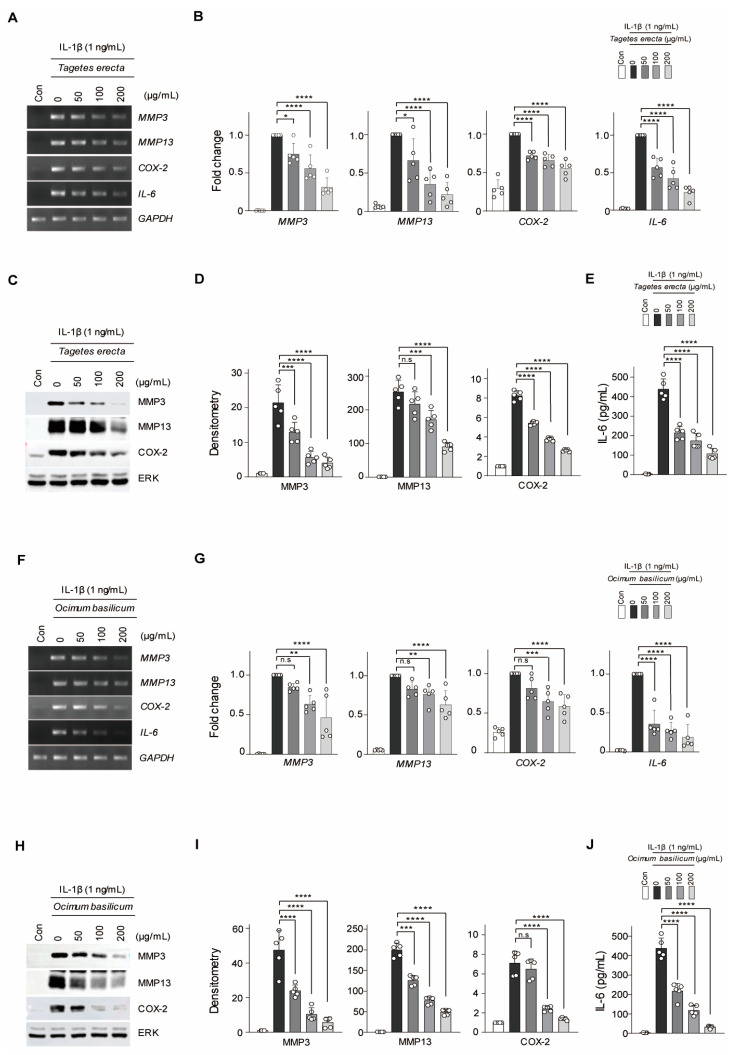
MMP3, MMP13, COX-2, and IL-6 were downregulated by *Tagetes erecta* and *O. basilicum* when co-treated with IL-1β in mouse articular chondrocytes. *T. erecta* or *O. basilicum* (50, 100, and 200 μg/mL) were used for co-treatment with IL-1β (1 ng/mL) in chondrocytes. (**A**,**B**,**F**,**G**) The mRNA relative expression levels of MMP3, MMP13, COX-2, and IL-6 were evaluated via reverse transcription–polymerase chain reaction (RT-PCR) and quantified by qRT-PCR (*n* = 5). (**C**,**D**,**H**,**I**) Protein levels of MMP3, MMP13, and COX-2 were detected using Western blot analysis and quantified by measuring relative intensities using densitometry. (**E**,**J**) IL-6 production was measured using ELISA. Transcription and protein levels were normalized to those of GAPDH or ERK, respectively. Data are represented as mean ± standard deviation (SD) as results of analysis by one-way analysis of variance (ANOVA) with Dunnett’s multiple comparisons test (*n* = 5). * *p* < 0.05, ** *p* < 0.01, *** *p* < 0.001, **** *p* < 0.0001, and n.s = not significant.

**Figure 2 ijms-24-17459-f002:**
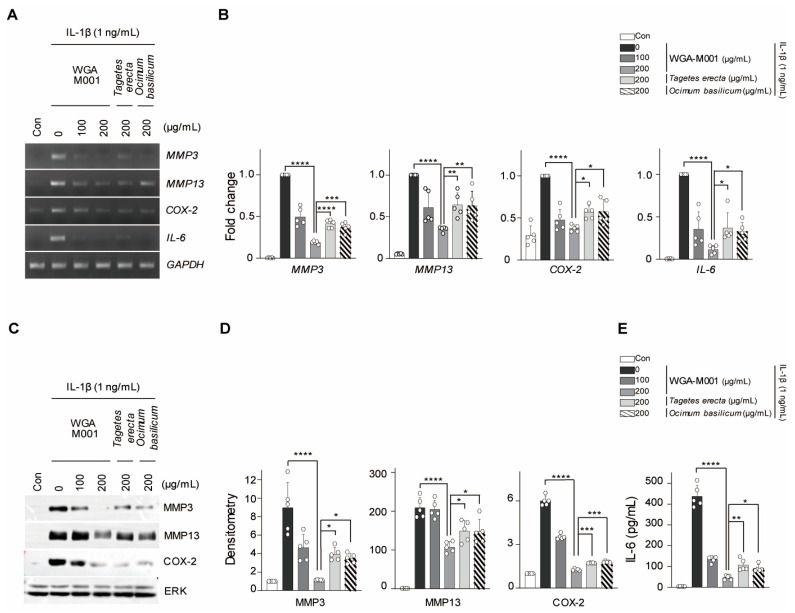
WGA-M001 was more effective than *Tagetes erecta* and *O. basilicum* alone. WGA-M001 (0, 100, and 200 μg/mL), *T. erecta* (200 μg/mL), or *O. basilicum* (200 μg/mL) were used to treat chondrocytes for 12 h with IL-1β (1 ng/mL). (**A**,**B**) Relative mRNA expression levels of MMP3, MMP13, COX-2, and IL-6 were evaluated using RT-PCR and quantified using qRT-PCR (*n* = 5). (**C**,**D**) Protein levels of MMP3, MMP13, and COX-2 were detected by Western blot analysis and relative intensities were quantified using densitometry. (**E**) IL-6 levels were measured using ELISA. Data are represented as mean ± SD as results of analysis by one-way ANOVA with Dunnett’s multiple comparisons test (*n* = 5). Transcription and protein levels were normalized to those of the GAPDH and ERK, respectively. * *p* < 0.05, ** *p* < 0.01, *** *p* < 0.001, **** *p* < 0.0001, and n.s = not significant.

**Figure 3 ijms-24-17459-f003:**
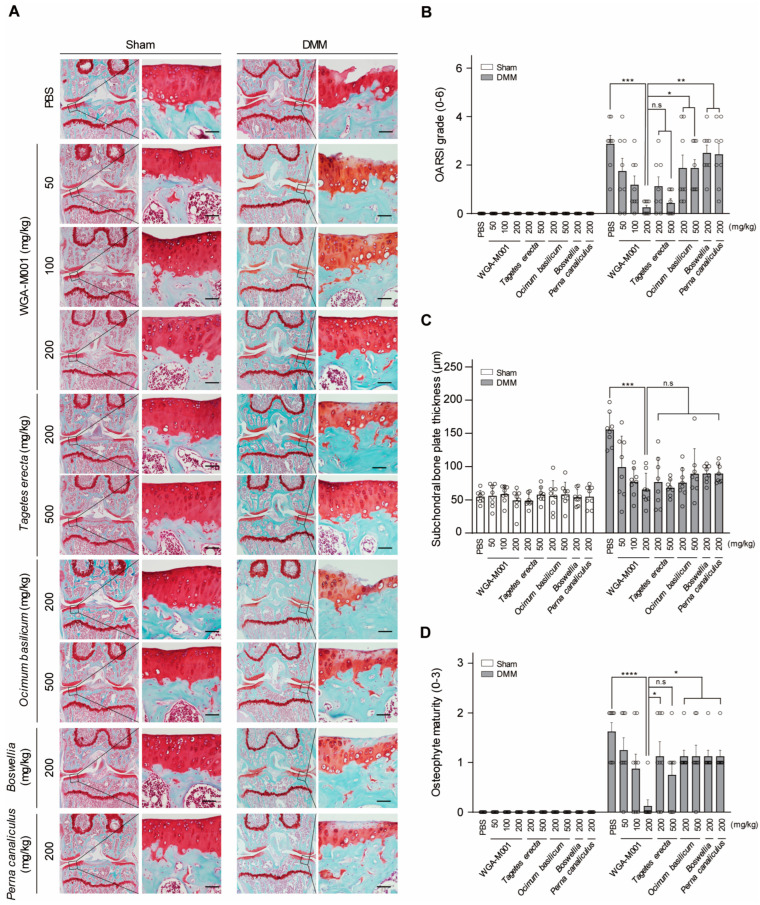
Oral administration of WGA-M001 reduced cartilage destruction in the DMM-induced OA mouse model (*n* = 8). Mice were treated with WGA-M001 (0, 50, 100, and 200 mg/kg), *T. erecta* (200 and 500 mg/kg), and *O. basilicum* (200 and 500 mg/kg) every day for 10 weeks. (**A**) The degree of cartilage destruction was analyzed using Safranin O staining (scale bar = 100 μm). To evaluate the degree of OA symptoms, OARSI grade (**B**), osteophyte maturity (**C**), and subchondral bone plate (**D**) were measured. Values are presented as mean ± SD and were assessed using the Kruskal–Wallis test followed by Dunn’s multiple comparisons test. * *p* < 0.05, ** *p* < 0.01, *** *p* < 0.001, **** *p* < 0.0001, and n.s = not significant.

**Figure 4 ijms-24-17459-f004:**
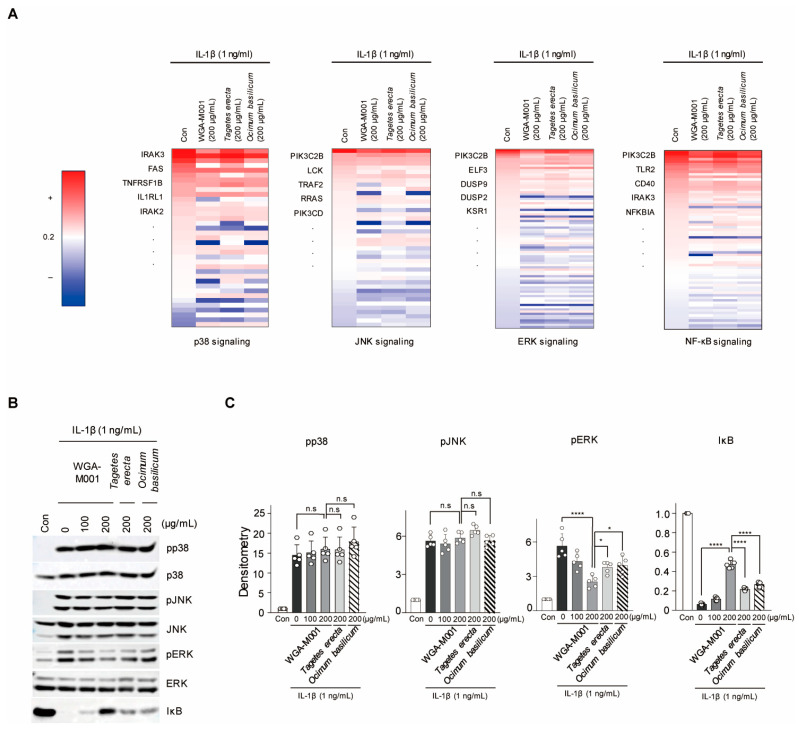
WGA-M001 regulated OA-related molecules through dephosphorylation in the NF-κB and ERK pathways. For in silico analysis, 200 μg/mL of *Tagetes erecta*, *Ocimum basilicum,* or WGA-M001 was used for treatment of chondrocytes for 12 h along with IL-1β (1 ng/mL), and RNA sequencing was performed. (**A**) In each signaling pathway, the number of genes upregulated by IL-1β and then downregulated by *T. erecta*, *O. basilicum,* or WGA-M001 was shown. (**B**,**C**) Protein levels of pp38, p38, pJNK, JNK, pERK, ERK, pp65, p65, and IκB were detected by Western blot analysis and relative intensities were quantified by densitometry (*n* = 5). Each protein level was normalized to ERK. Data are represented as mean ± SD as results of analysis by one-way ANOVA with Dunnett’s multiple comparisons test (*n* = 5). * *p* < 0.05, **** *p* < 0.0001, and n.s = not significant.

**Figure 5 ijms-24-17459-f005:**
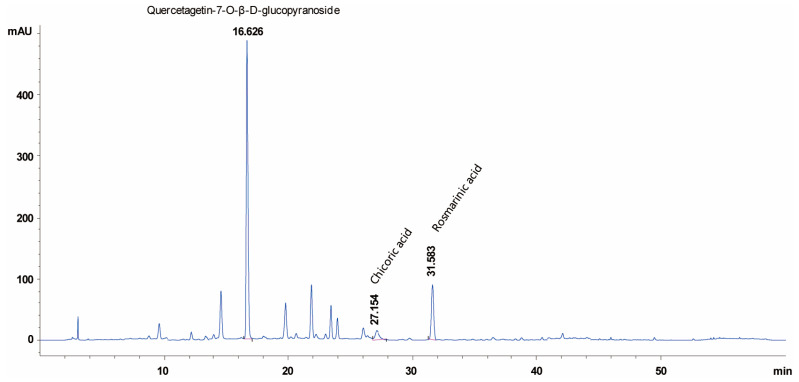
HPLC chromatogram of *WGA-M001*. Retention times of Quercetagetin-7-*O*-*β*-D-glucopyranoside, Chicoric acid, and Rosmarinic acid were 16.6, 27.1, and 31.5 min, respectively.

**Figure 6 ijms-24-17459-f006:**
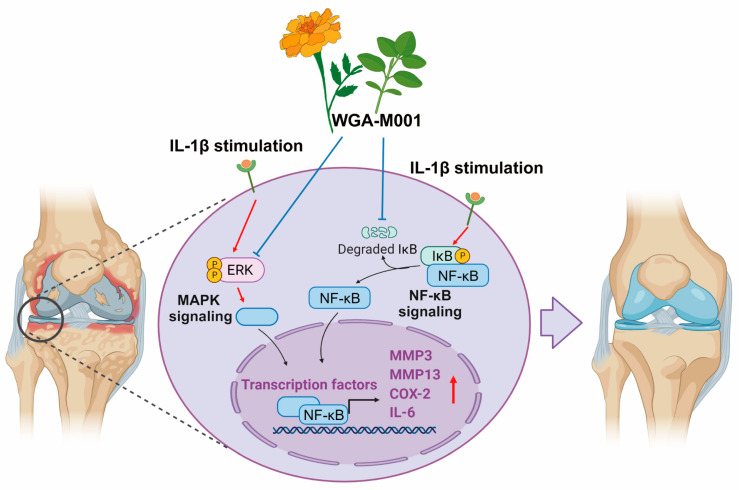
The graphical summary of WGA-M001 illustrates its role in alleviating OA-related catabolic molecules through the inhibition mechanism of ERK and NF-κB signaling pathways.

## Data Availability

The data used to support the findings of this study are available in the article and Appendix A.

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
