# Peer review of "WGA-M001, a Mixture of Total Extracts of *Tagetes erecta* and *Ocimum basilicum*, Synergistically Alleviates Cartilage Destruction by Inhibiting ERK and NF-κB Signaling"

_ijms, 2023, doi:10.3390/ijms242417459_

Round 1
Reviewer 1 Report
Comments and Suggestions for Authors
The manuscript titled " WGA-M001, a mixture of total extracts of Tagetes erecta and Ocimum basilicum, synergistically alleviates cartilage destruction by inhibiting ERK and NF-κB signaling " by Eunjeong Oh et al., aims to evaluate the effects of T. erecta, O. basilicum, and their mixture, WGA-M001, on in vitro and in vivo OA model systems.
Overall, the manuscript is well-written and provides scientifically significant information. The introduction is comprehensive and includes the main aspects of the study. The Materials and Method are written well giving the opportunity for reproduction of the experiments.
There are some issues which can be addressed to the authors:
- The next sentence ”T. erecta (200 μg/mL), O. basilicum (200 μg/mL), or WGA-M001 (200 μg/mL) were treated in IL-1β (1 ng/ml) treated mouse chondrocytes” is not written correctly. Maybe it could be “……………………………… were tested in pre-treated with IL-1β (1 ng/ml) mouse chondrocytes”
- The description of the pre-treatment of chondrocytes with IL-1β is missing.
- It is not mentioned the number of mice per group used in the experiments.
- Rows 98-99 - The sentence is not clearly written.
- It is not clear whether the chondrocytes were pre-treated with IL-1β, and subsequently, the effect of the studied substances was analyzed, or the cells were co-treated with IL-1β and the extracts. See Figure 1 text.
- Row 123 - The sentence is inexact.
- The text in Figure 3 and Figure 4 is too small to be read.
- The theory for the advantageous inhibitory effect of the WGA-M001 mixture is missing in the Discussion. Is it because of the presence of biologically active compounds like Quercetagetin-7-O-β-D-glucopyranoside, Chicoric acid and Rosmarinic acid, or something else? What is the presence of these compounds in the individual extracts?
Author Response
Reviewer #1:
Comments to the Author
The manuscript titled " WGA-M001, a mixture of total extracts of Tagetes erecta and Ocimum basilicum, synergistically alleviates cartilage destruction by inhibiting ERK and NF-κB signaling " by Eunjeong Oh et al., aims to evaluate the effects of T. erecta, O. basilicum, and their mixture, WGA-M001, on in vitro and in vivo OA model systems.
Overall, the manuscript is well-written and provides scientifically significant information. The introduction is comprehensive and includes the main aspects of the study. The Materials and Method are written well giving the opportunity for reproduction of the experiments.
There are some issues which can be addressed to the authors:
- The next sentence “T. erecta (200 μg/mL), O. basilicum (200 μg/mL), or WGA-M001 (200 μg/mL) were treated in IL-1β (1 ng/ml) treated mouse chondrocytes” is not written correctly. Maybe it could be “……………………………… were tested in pre-treated with IL-1β (1 ng/ml) mouse chondrocytes”
Author response and action:
We have addressed this point in the ‘Materials and Methods’ section of the revised manuscript as follows: T. erecta (200 μg/mL), O. basilicum (200 μg/mL), or WGA-M001 (200 μg/mL) were used for co-treatment with IL-1β (1 ng/ml) in the mouse chondrocytes.
- The description of the pre-treatment of chondrocytes with IL-1β is missing.
Author response and action:
No pretreatment analysis was performed. As mentioned above, we used IL-1β (1 ng/ml) along with T. erecta (200 μg/mL), O. basilicum (200 μg/mL), or WGA-M001 (200 μg/mL) for co-treatment. However, to improve our research quality, we have emphasized on this point in ‘sample treatment and reagents’ section of revised manuscript again.
“the samples (50, 100, and 200 μg/mL) were used along with IL-1β for the co-treatment of mouse articular chondrocytes.”
- It is not mentioned the number of mice per group used in the experiments.
Author response and action:
Each experiment was conducted independently at least five times. We have addressed this point in the revised manuscript.
- Rows 98-99 - The sentence is not clearly written.
Author response and action:
We have rewritten the sentence for clarity in revised manuscript.
“Herein, we demonstrate that WGA-M001, a mixture of T. erecta and O. basilicum extracts, strongly attenuates OA development through the inhibition of gene signatures related to ERK and NF-kB pathway activation, compared to the attenuation provided by a single extract of T. erecta or O. basilicum. Moreover, the effect of WGA-M001 strongly suggests the possibility of OA treatment in vitro and in vivo.”
- It is not clear whether the chondrocytes were pre-treated with IL-1β, and subsequently, the effect of the studied substances was analyzed, or the cells were co-treated with IL-1β and the extracts. See Figure 1 text.
Author response and action:
As mentioned above, each sample was co-treated with IL-1b in chondrocytes. We have addressed this point in Figure 1 in the revised manuscript.
- Row 123 - The sentence is inexact.
Author response and action:
Thank you for suggestion. We have changed this point in the revised manuscript.
“WGA-M001 (100 μg/ml and 200 μg/ml) was used with IL-1β as co-treatment of mouse chondrocytes.”
- The text in Figure 3 and Figure 4 is too small to be read.
Author response and action:
We apologize for the illegible text . We have increased the font size in all figures according to the MDPI guidelines. We have also uploaded each figure individually into the manuscript submission system of the International Journal of Molecular Sciences.
- The theory for the advantageous inhibitory effect of the WGA-M001 mixture is missing in the Discussion. Is it because of the presence of biologically active compounds like Quercetagetin-7-O-β-D-glucopyranoside, Chicoric acid and Rosmarinic acid, or something else? What is the presence of these compounds in the individual extracts?
Author response and action:
Thank you for your valuable comments. When we analyzed T. erecta and O. basilicum by HPLC analysis, respectively, Quercetagetin-7-O-β-D-glucopyranoside mainly existed in T. erecta and Chicoric acid, and Rosmarinic acid were found in O. basilicum (Data was not shown). However, WGA-M001 contained Quercetagetin-7-O-β-D-glucopyranoside, Chicoric acid, and Rosmarinic acid. The ratio of T. erecta to O. basilicum was 10:1. This combination may show a stronger inhibitory effect on WGA-M001 compared to that of T. erecta and O. basilicum total extracts. The HPLC data for WGA-M001 showed several peaks in a time-dependent manner. Although we have not characterized all the single compounds, they could be valuable single compounds with anti-catabolic or anti-osteoarthritic effects. In the near future, we will isolate each compound and characterize its anti-catabolic and anti-osteoarthritic effects. We have described this in the Discussion section of the revised manuscript.
We have attached the HPLC data and have shown only the reviewer because of next experiments.

Reviewer 2 Report
Comments and Suggestions for Authors
Figure 1: Why were different marker sets investigated (MMP3, MMP13, COX-2, IL-6) vs (MMP3, MMP13, COX-2, ERK)? The figure legend does not list all investigated markers, COX-2 or ERK are missing for mRNA and protein quantification, respectively. Data for MMP3 in Fig 1b and G seems very high and the presented gel images in A and F make it hard to believe that the fold change is more than 200-fold. Was “None” used to normalize expression? If yes, IL1beta-stimulated and 0 µg extract treated cells fit better as reference for quantification than unstimulated cells.
Line 123: sentence should read: “IL-1β-treated chondrocytes were treated with WGA-M001 in a dose-dependent manner”
Section 2.2.: The sentence “Interestingly, suppression of catabolic factor expression by WGA-M001 was stronger than T. erecta or O. basilicum in IL-1β treated chondrocytes.” Is very shallow and is not supported by the data presented in Figure 2. The results of statistical tests performed are not properly reported, the authors should indicate the value of the test statistic and degrees of freedom, as well as which condition is compared to what together with the respective result. The authors should also make sure to only plot relevant statistical differences, eg. difference between “0” and “None” is neither mentioned in the text, nor relevant at all as treatment effect of the extracts and not the effect of IL1beta stimulation is investigated. To make a claim that WGA-M001 is superior over single extract treatment, the three conditions involving 200µg need to be compared, actually nothing else.
Line 153: What is SBP (acronyms should be explained upon first use) or “osteophyte maturity”?
Line 162: The claim on “synergy” cannot be made on these data, please refer to literature on investigation of drug synergy and perform proper experiments: https://www.ncbi.nlm.nih.gov/pmc/articles/PMC7010330/, https://www.ncbi.nlm.nih.gov/pmc/articles/PMC7986484/
Figure 3 B-D: Statistically testing each condition against PBS is nonsense. If the authors want to conclude that WGA-M001 is superior over single extract treatments, those conditions should be compared – and the statistical test results properly reported (eg. “U(degrees of freedom)=.., p=..”). The current data presentation and unspecific description does not allow a conclusion that WGA-M001 is superior over other treatments.
Line 175: Synergistic or synergetic? Neither one is appropriate to describe the results, see comment above.
Figure 4: the font sizes are too small, a microscope would be needed to read the figure.
Figure 6: Please increase the size of the figure, the text is too small.
Line 240-247: This section is not relevant to the study, which investigated T. erecta, O. basilicum and WGA-M001.
Line 248: The sentence “Therefore, further research is needed to identify natural substances that are effective 248 against OA and to elucidate their mechanisms of action.” is irrelevant here.
Section 4.1: It should be described how the “50% ethanol extract powder” of each plant was prepared. Otherwise the study cannot be replicated independently by other researchers.
Line 343: “was isolated by RNA sequencing” – What does this mean?
Comments on the Quality of English LanguageModerate editing of English language required
Author Response
Reviewer #2:
Comments to the Author
- Figure 1: Why were different marker sets investigated (MMP3, MMP13, COX-2, IL-6) vs (MMP3, MMP13, COX-2, ERK)? The figure legend does not list all investigated markers, COX-2 or ERK are missing for mRNA and protein quantification, respectively. Data for MMP3 in Fig 1b and G seems very high and the presented gel images in A and F make it hard to believe that the fold change is more than 200-fold. Was “None” used to normalize expression? If yes, IL1beta-stimulated and 0 µg extract treated cells fit better as reference for quantification than unstimulated cells.
Author response and action:
We analyzed transcription and protein levels using PCR and western blot analyses, respectively. To check transcription levels, GAPDH was used as a housekeeping gene. For western blot analysis, ERK was used as a housekeeping gene. To determine the target gene or protein expression levels by qRT-PCR and densitometry, respectively, MMP3, MMP13, and COX-2 were normalized to GAPDH or ERK. We described this in the figure legends.
We have added information on COX-2 to the figure legend. We also repeated densitometry analysis and corrected the graph in the revised manuscript.
We had mistyped the word ‘None’. This was used as the control. We used it as a control because all samples were dissolved in DMSO. We have modified and described this in the revised figure legends.
- Line 123: sentence should read: “IL-1β-treated chondrocytes were treated with WGA-M001 in a dose-dependent manner”
Section 2.2.: The sentence “Interestingly, suppression of catabolic factor expression by WGA-M001 was stronger than T. erecta or O. basilicum in IL-1β treated chondrocytes.” Is very shallow and is not supported by the data presented in Figure 2. The results of statistical tests performed are not properly reported, the authors should indicate the value of the test statistic and degrees of freedom, as well as which condition is compared to what together with the respective result. The authors should also make sure to only plot relevant statistical differences, eg. difference between “0” and “None” is neither mentioned in the text, nor relevant at all as treatment effect of the extracts and not the effect of IL1beta stimulation is investigated. To make a claim that WGA-M001 is superior over single extract treatment, the three conditions involving 200µg need to be compared, actually nothing else.
Author response and action:
As mentioned above, the word “None” was originally intended to denote controls. We have described the data analysis in detail in the revised manuscript.
“The expression level of MMP3, MMP13, COX-2, and IL-6 increased with the IL-1β treatment, compared to the levels in the control, whereas IL-1β induced catabolic factors decreased in the presence of WGA-M001 in a dose-dependent manner. Moreover, the decrease in the level of catabolic factor caused by WGA-M001 (200 μg/ml) was greater than the decrease associated with the same concentration of single extract T. erecta or O. basilicum in chondrocytes.”
- Line 153: What is SBP (acronyms should be explained upon first use) or “osteophyte maturity”?
Author response and action:
To determine the severity of OA development, we examined cartilage destruction using subchondral bone plates thickness and osteophyte maturity [35]. The SBP plays an important role in articular cartilage metabolism [36]. Osteophyte maturity indicates the amount of bone tissue in osteophytes [37]. OA is a whole-joint disease characterized by increased cartilage destruction, osteophyte formation, and subchondral bone plates thickness [38]. This is described in the revised manuscript.
“References
- Wei, Y.; Luo, L.; Gui, T.; Yu, F.; Yan, L.; Yao, L.; Zhong, L.; Yu, W.; Han, B.; Patel, J. M.; Liu, J. F.; Beier, F.; Levin, L. S.; Nelson, C.; Shao, Z.; Han, L.; Mauck, R. L.; Tsourkas, A.; Ahn, J.; Cheng, Z.; Qin, L., Targeting cartilage EGFR pathway for osteoarthritis treatment. Sci Transl Med 2021, 13, (576).
- Madry, H.; van Dijk, C. N.; Mueller-Gerbl, M., The basic science of the subchondral bone. Knee Surg Sports Traumatol Arthrosc 2010, 18, (4), 419-33.
- van der Kraan, P. M.; van den Berg, W. B., Osteophytes: relevance and biology. Osteoarthritis Cartilage 2007, 15, (3), 237-44.
- Jeon, J.; Noh, H. J.; Lee, H.; Park, H. H.; Ha, Y. J.; Park, S. H.; Lee, H.; Kim, S. J.; Kang, H. C.; Eyun, S. I.; Yang, S.; Kim, Y. S., TRIM24-RIP3 axis perturbation accelerates osteoarthritis pathogenesis. Ann Rheum Dis 2020, 79, (12), 1635-1643.”
- Line 162: The claim on “synergy” cannot be made on these data, please refer to literature on investigation of drug synergy and perform proper experiments: https://www.ncbi.nlm.nih.gov/pmc/articles/PMC7010330/, https://www.ncbi.nlm.nih.gov/pmc/articles/PMC7986484/
Author response and action:
Unfortunately, the limited time provided for this revision was insufficient to determine a synergistic effect. We have changed the term synergy effect to combination effect. This information has been included in the revised manuscript.
- Figure 3 B-D: Statistically testing each condition against PBS is nonsense. If the authors want to conclude that WGA-M001 is superior over single extract treatments, those conditions should be compared – and the statistical test results properly reported (eg. “U(degrees of freedom)=.., p=..”). The current data presentation and unspecific description does not allow a conclusion that WGA-M001 is superior over other treatments.
Author response and action:
To improve our explanation of the data, we have described this in greater detail in the revised manuscript. WGA-M001 had an anti-osteoarthritic effect compared to the effect of the oral administration of PBS as a control in DMM-induced OA mice. Furthermore, the oral administration of a small amount of WGA-M001 (up to 100 mg/kg) protected against cartilage destruction, whereas high doses (500 mg/kg) of T. erecta and O. basilicum exerted anti-osteoarthritic effects. If humans show anti-osteoarthritic effects after oral administration of T. erecta or O. basilicum extracts, they should be administered at a daily dose of 87.5 mg/kg. Therefore, WGA-M001 was more attractive for OA treatment than the other single extracts were. As your concerns, we also checked the statistical analysis again and performed a post hoc test using Dunn’s multiple comparisons test instead of Mann-Whitney U test. This has been described in the revised manuscript.
- Line 175: Synergistic or synergetic? Neither one is appropriate to describe the results, see comment above.
Author response and action:
As mentioned above, we have changed the term “synergy effect” to “combination effect.”
- Figure 4: the font sizes are too small, a microscope would be needed to read the figure.
Author response and action:
We apologize for the illegible text. We have increased the font size in all figures according to the MDPI guidelines. We have also uploaded each figure individually to the manuscript submission system of the International Journal of Molecular Sciences.
- Figure 6: Please increase the size of the figure, the text is too small.
Author response and action:
We agree with the reviewer’s suggestions. The font size has been increased. This information has been included in the revised manuscript.
- Line 240-247: This section is not relevant to the study, which investigated T. erecta, O. basilicum and WGA-M001.
Author response and action:
Accordingly, we have deleted lines 240–247. Instead of deleting lines or sentences, we have added the HPLC data of T. erecta and O. basilicum, as suggested by reviewer #1.
- Line 248: The sentence “Therefore, further research is needed to identify natural substances that are effective 248 against OA and to elucidate their mechanisms of action.” is irrelevant here.
Author response and action:
In accordance with your suggestion, we have deleted line 248.
- Section 4.1: It should be described how the “50% ethanol extract powder” of each plant was prepared. Otherwise, the study cannot be replicated independently by other researchers.
Author response and action:
Thank you for your comment. We have described this point in greater detail in the revised manuscript.
“Dried T. erecta and O. basilicum was extracted using 50% EtOH for 24 h at 25°C. The extracts were filtered using a Buchner funnel and concentrated using a rotary evaporator to obtain a 50% EtOH extraction powder. The 50% ethanol extract was fractionated using ethyl acetate and concentrated using a rotary evaporator. The dried samples were then stored at − 80°C in the refrigerator until further use.”
- Line 343: “was isolated by RNA sequencing” – What does this mean?
Author response and action:
We apologize for this typographical error. RNA sequencing was performed using TRIzol reagent (Molecular Research Center, Cincinnati, OH, USA). This information has been included in the revised manuscript.
“Total RNA was isolated from mouse chondrocytes using TRIzol reagent (Molecular Research Center Inc., Cincinnati, OH, USA)”

Round 2
Reviewer 2 Report
Comments and Suggestions for Authors
Dear authors,
Thank you for addressing most of the comments, however, some substantial issues remain:
Figure 1: The authors seem to confuse the terms “relative expression” and “fold change” – they are not synonyms. Currently, the authors present the data as relative expression. More interesting though would be fold change with respect to the 0 µg (“untreated”) condition. Please calculate fold changes, adapt the figure accordingly and perform the proper statistical testing (here, multiple comparison using Tukey post hoc in graph pad would make sense). The same applies to figure 2.
Line 130: “one way” is repeated.
Line 175/176: “Oral administration of a small amount of WGA-M001 (up to 100 mg/kg) provided protection against cartilage destruction, whereas high doses (500 mg/kg) of T. erecta and O. basilicum showed anti-osteoarthritic effects.” – What is the difference between “cartilage destruction” and “anti-osteoarthritic effect”? This wording makes a reader think, the different treatments have completely different mechanisms of action.
Line 177/178: “If humans show anti-osteoarthritic effects after oral administration of T. erecta or O. basilicum extracts, they should be administered at a daily dose of 87.5 mg/kg.” – Which data support this claim?
In general, the authors still did not manage to correctly statistically compare individual treatment condtions which are reported on in the text. Eg. line 186: “WGA-M001 had a greater protective effect against OA development than the effect of the oral administration of Boswellia and P. canaliculus.” – Figure 3 still compares each condition against PBS. However, to make a statement about efficacy of one treatment condition over another, those two conditions have to be compared, and not individually against PBS, which is nonsense. This applies to almost all comparisons made in the manuscript.
Author Response
Reviewer #2:
Comments to the Author
Thank you for addressing most of the comments, however, some substantial issues remain:
- Figure 1: The authors seem to confuse the terms “relative expression” and “fold change” – they are not synonyms. Currently, the authors present the data as relative expression. More interesting though would be fold change with respect to the 0 µg (“untreated”) condition. Please calculate fold changes, adapt the figure accordingly and perform the proper statistical testing (here, multiple comparison using Tukey post hoc in graph pad would make sense). The same applies to figure 2.
Author response and action:
Thank you for your valuable comments. We change Relative expression to fold change. Indeed, we analysis Data of Figure 1, 2, and 4 with Tukey post hoc. You can find this point in revised Figure 1, 2, and 4 of main text.
- Line 130: “one way” is repeated.
Author response and action:
We apologize for this typographical error. We deleted repeated word in revised manuscript.
“ ~ ~as results of analysis by one-way analysis of variance (ANOVA) with Tukey post‐hoc test (n = 5).”
- Line 175/176: “Oral administration of a small amount of WGA-M001 (up to 100 mg/kg) provided protection against cartilage destruction, whereas high doses (500 mg/kg) of T. erecta and O. basilicum showed anti-osteoarthritic effects.” – What is the difference between “cartilage destruction” and “anti-osteoarthritic effect”? This wording makes a reader think, the different treatments have completely different mechanisms of action.
Author response and action:
We agree your suggestion. We remove anti-osteoarthritic effects and change suppression of cartilage destruction in revised manuscript.
“,whereas high doses (500 mg/kg) of T. erecta and O. basilicum showed suppression of cartilage destruction.”
- Line 177/178: “If humans show anti-osteoarthritic effects after oral administration of T. erecta or O. basilicum extracts, they should be administered at a daily dose of 87.5 mg/kg.” – Which data support this claim?
Author response and action:
In accordance with your suggestion, we have deleted line 177/178. You can find this point in revised manuscript.
- In general, the authors still did not manage to correctly statistically compare individual treatment condtions which are reported on in the text. Eg. line 186: “WGA-M001 had a greater protective effect against OA development than the effect of the oral administration of Boswellia and P. canaliculus.” – Figure 3 still compares each condition against PBS. However, to make a statement about efficacy of one treatment condition over another, those two conditions have to be compared, and not individually against PBS, which is nonsense. This applies to almost all comparisons made in the manuscript.
Author response and action:
As your comment, we compared WGA-M001 vs other single compound after PBS + DMM vs WGA-M001 + DMM with Kruskal–Wallis test followed by the Dunn’s multiple comparisons test, respectively. After this statical analysis, we found that amount of WGA-M001 oral administration is at least fine time less than that of the T. erecta or O. basilicum extracts for suppressing cartilage destruction. Indeed, we change the sentence “WGA-M001 had a greater protective effect against OA development than the effect of the oral administration of Boswellia and P. canaliculus.” To “WGA-M001 had a protective effect against cartilage destruction, compared to Boswellia and P. canaliculus.”. You can find this pint in revised manuscript.
“The amount of WGA-M001 oral administration is at least fine time less than that of the T. erecta or O. basilicum extracts for suppressing cartilage destruction”
“WGA-M001 had a protective effect against cartilage destruction, compared to Boswellia and P. canaliculus.”

Round 3
Reviewer 2 Report
Comments and Suggestions for Authors
Dear authors,
The reviewer complains about the following ignorant change made by authors: Axis labels of Figure 1 and 2 were simply changed from relative expression to fold change, without calculating actual fold change, as the values are still the same in the figure. The reviewer explicitly highlighted that “relative expression” to “fold change” are not synonyms, however the authors treated them as synonym, which is highly unscientific. Please calculate fold changes in relation to “0 µg/ml treated” condition. Condition “Con” can be neglected in the calculation, as comparing against absent expression results in nonsense values for the treatment conditions. If this concept goes beyond the understanding of the authors, please consult a scientific advisor. In addition, the reviewer wonders why all statistical comparisons are referencing “0 µg/ml treated” condition (significant or not), but not even n.s. differences are shown between the other treatment conditions. Therefore, it is doubted whether really Tukey’s post hoc test was performed, as the shown results are typical of a Dunetts test comparing against the “0 µg/ml treated” condition. Comparing against the “0 µg/ml treated” condition is correct, however, the reporting could be improved. Insignificant comparisons can be omitted. Again, comparing conditions “Con” and “0” is irrelevant here and can be omitted.
Line 152: There is an orphan “€” sign, maybe autocorrection by Word changin (E) to €?
Figure 4: On line 200, the authors state that “To demonstrate that WGA-M001 has a combined effect, its molecular mechanism was characterized in comparison with those of T. erecta and O. basilicum.” – The data of the performed RNA-seq analysis was only qualitatively analyzed with regard to up- or downregulation of a selected list of target genes. To be able to claim a “combined effect”, a quantitative expression analysis of individual target genes in the context of the whole population is required, which would indicate to which extent a given signaling pathway might be regulated by a given treatment in comparison to another. A comparison of the number of regulated genes is not appropriate here to support the claim of a combined effect. The presented Western Blot and its densitometric quantification outline that pERK and IkB are indeed regulated by the treatments, however, WGA-M001 at 200 µg/ml seems not to be better than either single compound at regulating ERK. The mean values were compared to untreated, however, it is essential to also compare Tagetes erecta, Ocimum basilicum and WGA-M001 at 200 µg/ml in Figure 4C (pERK) to be able to claim a combined effect. Based on the presented data in figure 4, this claim cannot be made.
The reviewer is still awaiting the proper reporting of statistical test results of mean comparisons in the main text rather than descriptive sentences like “The amount of WGA-M001 oral administration is at least fine time less than that of the T. erecta or O. basilicum extracts for suppressing cartilage destruction” (What does “least fine time” mean here?) If “five times” is meant – five times what, on average?
Comments on the Quality of English LanguageMinor issues present, pluralisation
Author Response
Reviewer #2:
Comments to the Author
Thank you for addressing most of the comments, however, some substantial issues remain:
- The reviewer complains about the following ignorant change made by authors: Axis labels of Figure 1 and 2 were simply changed from relative expression to fold change, without calculating actual fold change, as the values are still the same in the figure. The reviewer explicitly highlighted that “relative expression” to “fold change” are not synonyms, however the authors treated them as synonym, which is highly unscientific. Please calculate fold changes in relation to “0 µg/ml treated” condition. Condition “Con” can be neglected in the calculation, as comparing against absent expression results in nonsense values for the treatment conditions. If this concept goes beyond the understanding of the authors, please consult a scientific advisor. In addition, the reviewer wonders why all statistical comparisons are referencing “0 µg/ml treated” condition (significant or not), but not even n.s. differences are shown between the other treatment conditions. Therefore, it is doubted whether really Tukey’s post hoc test was performed, as the shown results are typical of a Dunetts test comparing against the “0 µg/ml treated” condition. Comparing against the “0 µg/ml treated” condition is correct, however, the reporting could be improved. Insignificant comparisons can be omitted. Again, comparing conditions “Con” and “0” is irrelevant here and can be omitted.
Author response and action:
We firstly check whether T. erecta and O. basilicum have suppression effect of catabolic factor expression in Figure 1 before analysing WGA-M001. We also compared 0 µg/ml treatment vs different dose treatment of T. erecta and O. basilicum with one-way analysis of variance (ANOVA) with Dunnett’s multiple comparisons test.
As you recognized in figure 2 and 4, we proved WGA-M001 has a strong suppression effect of catabolic factor expression through inhibition of ERK and NF-κB pathway, compared to T. erecta and O. basilicum extract treatment. To prove this hypothesis, we recalculate our data with one-way analysis of variance (ANOVA) with Dunnett’s multiple comparisons test. We describe this point in revised manuscript.
In figure 2: “This data collectively suggested that WGA-M001 have inhibitory effect of catabolic factor expression, compared to T. erecta and O. basilicum treatment.”
In figure 4: “WGA-M001 have inhibitory effect of catabolic factor expression, compared T. erecta and O. basilicum treatment through inhibition of ERK and NF-κB pathway.”
- Line 152: There is an orphan “€” sign, maybe autocorrection by Word changin (E) to €?
Author response and action:
We apologize for this typographical error. We corrected this point in revised manuscript.
“E”
- Figure 4: On line 200, the authors state that “To demonstrate that WGA-M001 has a combined effect, its molecular mechanism was characterized in comparison with those of T. erecta and O. basilicum.” – The data of the performed RNA-seq analysis was only qualitatively analyzed with regard to up- or downregulation of a selected list of target genes. To be able to claim a “combined effect”, a quantitative expression analysis of individual target genes in the context of the whole population is required, which would indicate to which extent a given signaling pathway might be regulated by a given treatment in comparison to another. A comparison of the number of regulated genes is not appropriate here to support the claim of a combined effect. The presented Western Blot and its densitometric quantification outline that pERK and IkB are indeed regulated by the treatments, however, WGA-M001 at 200 µg/ml seems not to be better than either single compound at regulating ERK. The mean values were compared to untreated, however, it is essential to also compare Tagetes erecta, Ocimum basilicum and WGA-M001 at 200 µg/ml in Figure 4C (pERK) to be able to claim a combined effect. Based on the presented data in figure 4, this claim cannot be made.
Author response and action:
We followed previous our publication with gene signature of NF-kB and MAP kinase signalling pathway. We already mentioned this point in Material & Method section. To check the WB data, we recalculate our data with one-way analysis of variance (ANOVA) with Dunnett’s multiple comparisons test. We describe this point in revised manuscript.
- The reviewer is still awaiting the proper reporting of statistical test results of mean comparisons in the main text rather than descriptive sentences like “The amount of WGA-M001 oral administration is at least fine time less than that of the T. erecta or O. basilicum extracts for suppressing cartilage destruction” (What does “least fine time” mean here?) If “five times” is meant – five times what, on average?
Author response and action:
We changed this sentence you and reader to easily understand what this means in revised manuscript.
“This shown that the amount of WGA-M001 oral administration is less than that of the T. erecta and O. basilicum extracts for suppressing cartilage destruction.”
